# ACID: A COMPREHENSIVE DATASET FOR AI-CREATED IMAGE DETECTION

## ABSTRACT

Generative models have demonstrated remarkable capabilities in generating photorealistic images under proper conditional guidance. Such advancements raise concerns about potential negative social impacts, such as the proliferation of fake news. In response, numerous methods have been developed to differentiate fake from real. Yet, their accuracy and reliability still need to be improved, especially when facing state-of-the-art generative models such as large diffusion models. Infrastructure-wise, the existing testing datasets are sub-optimal in terms of research dimensions and product utility due to their limited data volume and insufficient domain diversity. In this work, we introduce a comprehensive new dataset, namely ACID, which consists of 13M samples sourced from over 50 different generative models versus real-world scenarios. The AI-generated images in this collection are sampled based on fine-grained text prompts and span multiple resolutions. For the real-world samples, we broadly searched public data sources and carefully filtered text-image pairs based on visual and caption quality. Using ACID, we present ACIDNet, an effective framework for detecting AI-generated images. ACIDNet leverages texture features from a Single Simple Patch (SSP) branch and semantic features from a ResNeXt50 branch, and achieves overall cross-benchmark accuracy of $86.77\%$, significantly outperforming previous methods such as SSP and CNNSpot by over $10\%$. Both our model and dataset will be open-released to the public.

## 1 INTRODUCTION

Over the past few years, image generation has seen significant advancements due to the development of deep learning models such as VAEs [24], GANs [17], and diffusion-based systems [42]. While these technologies have greatly facilitated creative endeavors, they also pose serious societal risks, such as the generation of fake news through deepfake technologies [43]. Recently, the text-to-image generation models, i.e., diffusion-based models [42], have become widely adopted due to their impressive performance. However, this capability is also being exploited to create fake images that are increasingly difficult to distinguish from genuine ones. Consequently, there is an urgent need to develop effective methods for detecting AI-created images.

Numerical methods have been developed to detect AI-created images. While most detectors have focused on identifying images produced by GANs [52], PatchCraft [59] introduced a generalized detector capable of identifying GAN-based and diffusion-based images. This was further supported by testing on a subset of the dataset proposed in the paper as a benchmark that encompasses a diverse range of AI-created images. Additionally, SSP [7] implemented a pipeline that uses a single simple patch as a fingerprint, achieving improved results on this benchmark. However, these two methods still exhibit limitations for real-world application. Both are predicated on the same hypothesis: the noise will appear in the simple or background patches of the generated images affected by the CNN-based up-sampler [14]. This assumption may not always hold, potentially affecting their effectiveness in diverse scenarios. The underlying assumption of these methods is validated within their benchmark, which features generated images that are simplistic and of low quality, often containing numerous artifacts, as shown in Figure 1a. However, as generative models have evolved, recent advancements enable the production of images with more precise and richer content, conditioned through text. This progress underscores the need to develop more challenging datasets that are better aligned with practical applications and current technologies.

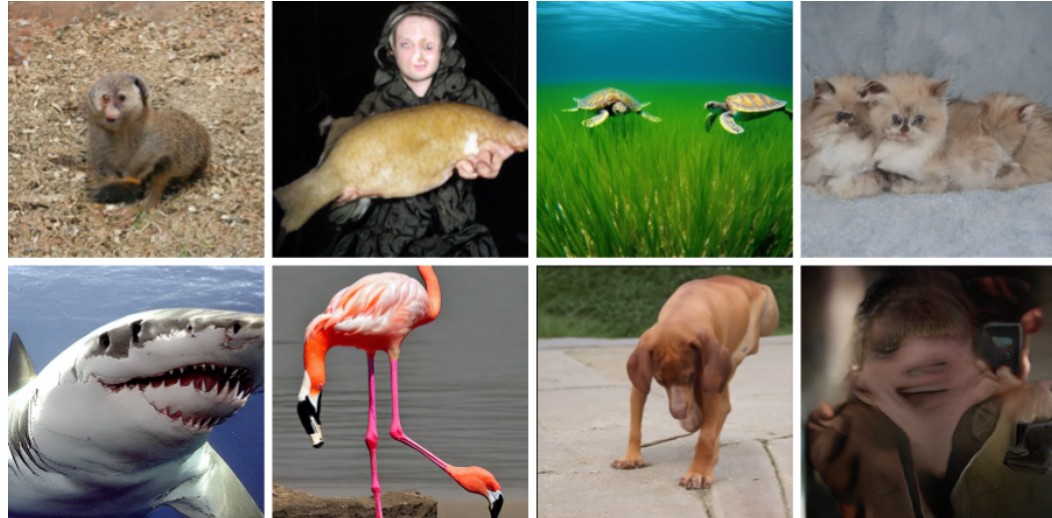

(a) Images of low significance: low quality and resolution, can be easily distinguished by humans.

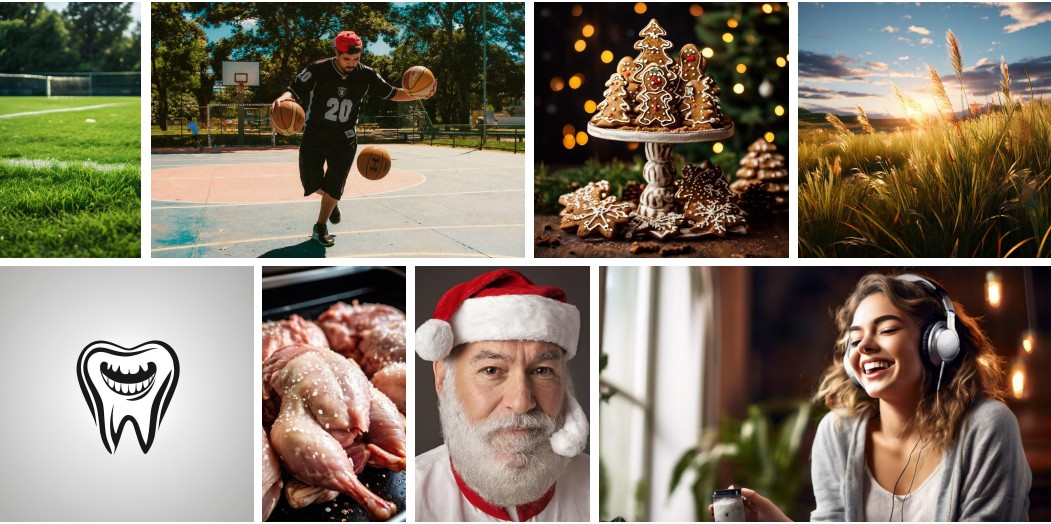

(b) Images of higher significance: only one image in this set is not synthetic. Can you identify it?

Figure 1: Given the challenges that AI-created content poses to content trustworthiness, we aim to focus more on images that are difficult for humans to distinguish. Besides, we are trying to make images contain content of greater value in practical applications.

To address current limitations, we propose a new dataset named ACID that includes a total of 13M authentic and AI-generated images, where over 95% of them have a resolution of at least $512 \times 512$. The 4.1M real-world images sourced from ImageNet-21K [12], the COCO dataset [26], and various free stock image repositories. Additionally, it contains 8.9M AI images either generated by single generative models such as Kandinsky2.2 [4], Kandinsky 3 [3], Realistic Vision 5.1 [2], Realism Engine 1.0 [41], Juggernaut XL V9 [45], RealVisXL 4.0 [49], Stable Diffusion XL [37], MidJourney V5/V6 [29], Dall-E 2/3 [40, 32], Playground 2.5 [36], Stable Cascade [34] and Photo Realistic [35], or by combinations of fine-tuned checkpoints and additional adapters from Civitai [10].

To develop a dataset that better aligns with practical applications, we propose a series of methods for both AI-created and real-world images, incorporating a broader range of models. For AI-created images, first, we employ meticulous prompt engineering to achieve realistic effects, examples of which are illustrated in Figure 1b. Second, we generate images at the recommended resolutions for each diffusion model to optimize performance. Finally, we collected images from Civitai [10]

that are posted by community users, of relatively higher quality, and closer to the real-life usage of generative models.

For a more complex real-world image dataset, we expand our collection beyond traditional photographs to include non-photorealistic images and those with minimal editing. This approach enhances the diversity and applicability of our benchmarking framework. Due to these efforts, among the state-of-the-art pretrained classifiers [52, 20, 28, 50, 31, 53, 7], the highest accuracy rate on our benchmark is only 65.65%, highlighting the necessity for a more robust baseline to improve recognition of AI-created images. Using human evaluations, we also compare our ACID and other benchmarks [59, 31]. These assessments demonstrated that ACID possesses a significantly greater capacity to confuse human observers.

Alongside an dataset, we introduce an effective and robust method named ACIDNet , which consists of two branches: a semantic branch and a texture branch. The semantic branch employs the ResNeXt50 model [55] to extract semantic features from the entirety of the input image. Inspired by the methodologies of SSP [7] and PatchCraft [59], the texture branch selects a patch exhibiting the lowest degree of pixel fluctuation as input. The texture and semantic features are then concatenated and processed through a fully connected layer for binary classification. Notably, ACIDNet achieves an accuracy rate of 98.01% on our benchmark of 100K images, representing a significant improvement of approximately 21.1% over the SSP method. We further validate the robustness of ACIDNet using the benchmark introduced in PatchCraft [59] and UnivFD [31]. On these benchmarks, ACIDNet consistently achieves an average accuracy of 81.1%, which is comparable to other state-of-the-art baselines.

To summarize, our contributions are as follows:

- We introduce a comprehensive new dataset called ACID consisting of 22M images, featuring high-quality outputs from over 50 distinct generative models alongside filtered real-world images. Over 95% of the images have a minimum resolution of $512 \times 512$ pixels.

- We have developed a series of methodologies to construct a dataset that includes prompt engineering for AI-created images, along with the collection of non-realistic and simply edited photos to represent real-world scenarios. According to human evaluation, the dataset with such methods poses a significant challenge in differentiating AI-created images from real-world images due to their enhanced quality and complexity.

- An effective and robustness baseline called ACIDNet is proposed in this work, which is composed of a semantic branch utilizing ResNeXt50 and a texture branch derived from SSP methodologies. We also have an ablation study to show the effectiveness of our ACIDNet.

- We have conducted extensive testing of various AI detectors on our dataset to demonstrate its challenging nature. Notably, our ACIDNet model achieves an impressive accuracy of 98.01% on this benchmark, marking a significant advancement in the recognition of AI-created images. We plan to release both the model weights and the dataset to the community, thereby contributing to advancements in AI safety.

## 2 RELATED WORKS

**Image generation research** has made significant progress in recent years. Many achieved to synthesize realistic-looking images using the well-known GAN pipeline [5, 9, 22, 21, 33] as well as the newly raised diffusion pipeline [37, 39, 40, 57, 48, 42, 56]. The inference of GAN models [5, 9, 22, 21, 33] requires a generator doing one feedforward pass conditioned on randomly sampled latent vectors. While the diffusion models [37, 39, 40, 57, 48, 42, 56] iteratively denoise normally sampled 2D noises (i.e. backward diffusion process) and synthesize real-looking images at the end of the loop. In recent years, the open community of CV has proposed a lot of high-performing diffusion models [4, 3, 2, 41, 45, 49, 36, 35] that gathered significant attention both in industry and research. As the goal of this work is to propose a reliable dataset for GenAI image detection, we therefore sampled synthetic images using the models listed above to achieve better quality and broader domains.

**AI-generated image detection** aims to determine whether an image is real or has been produced by generative models. Traditionally, this task is approached as a binary classification problem, where

simple CNN models often perform adequately if the training and testing images originate from the same generative model [52, 20, 53]. However, models such as UnivFD [31] that leverage large pretrained vision-language models for feature extraction tend to experience significant performance degradation when tested with images from previously unseen models [59]. To enhance the detection of images from new, unseen generative models, Fredect [15] introduced a method that detects generated images based on frequency anomalies. Additionally, LNP [27] employs a denoising model to extract noise patterns from images, while PatchCraft [59] focuses on the differential noise characteristics between foreground and background for better generalization. Building upon these ideas, SSP [7] simplifies the approach by using noise from a single, simple patch of the image. In this work, we propose a novel ACIDNet as our baseline based on SSP and CNN-based structures, which can obtain both texture features and semantic features.

# 3 ACID DATASET

## 3.1 MOTIVATION

Unlike traditional image classification tasks, the challenge of identifying AI-created images is dynamically evolving with advancements among generative models. According to our evaluation in Table 4, even a classifier with excellent generalization capabilities may struggle to recognize images produced by the latest diffusion models if only trained by early GANs. Therefore, an effective AI-created image detection dataset is supposed to contain a sufficiently diverse collection of authentic and AI-created images, along with the capacity to extend to future models easily.

In the datasets used in previous research like [31, 59, 61], the images are mainly in realistic (photographic) style, and each contains only one object from a specific category. This was primarily due to the limitations of the early generative models, predominantly those based on GANs. However, with the development of generative models, the demand for distinguishing AI-created images extends beyond the domains covered by these datasets. Consider the following scenarios:

- An individual collects public works of an artist and generates content closely resembling the artist's style, subsequently claiming it to be the artist's original work for profit.
- A forgery created with generative models fabricates content unattainable in reality, such as a tree concurrently bearing apples and watermelons. (Instances of AI-created images used to simulate forgery in academic publications are already documented.)
- A user captures a photo and enhances it by applying a filter or adding some text to improve its presentation before uploading.

A model developed based on a dataset as described above may struggle to identify the first two examples as AI-created images due to the discrepancies in their distributions, but mistakenly classify the last example as an AI-created image for its non-photographic elements. The potential challenges highlighted above necessitate the creation of a more comprehensive dataset with images meeting the following requirements: (1) contain various styles aside from photographic ones; (2) include rich content like scenes with no or multiple objects; (3) with higher and more diverse resolutions; (4) might have undergone primary, non-creative edits.

## 3.2 CONSTRUCTION

**Real-world images**     An important but often overlooked fact is that, to construct a robust dataset for AI-created image detection, collecting a diverse range of real-world images is just as crucial as gathering AI-created images. The real-world images in ACID are from the following sources:

- 349K images from ImageNet-21K dataset. For each label, only the samples with the highest resolution are retained.
- 287K images from the COCO dataset.
- 1.76M free stock images with captions of at least $512 \times 512$ resolution.
- We searched images posted on a large online platform A with over 10 million users, and selected 1.71M images that (1) were uploaded before 2019 and thus are not AI-created, and (2) have a history of user interactions, indicating relatively high quality.

Images from ImageNet-21K [12] and COCO [26] dataset aim to increase the content variance. Stock and online platform user images further diversify the dataset by introducing various styles and edited images, also making it more representative of what people typically see in real life.

**AI-created images**     The AI-created images in ACID are obtained from over 50 state-of-the-art generative image models in different ways depending on the nature of the models:

- For early models (mostly GAN-based ones) where the generative capacity is constrained to a limited variety of individual objects, the content and style of the generated images are bounded to their domains. We also introduce images from [52, 31, 59, 61] on such models.

- For open-sourced text-to-image models, images are generated using a list of carefully selected text prompts (see Table 1) at their recommended resolutions for best quality, and at two common aspect ratios to accommodate different use cases.

- For closed or commercial models, we have collected publicly released images [10, 51, 38] to the best of our ability.

- To further cover various fine-tuned checkpoints and combinations of adapters (Dreambooth [44], LoRA [19], ControlNet [58], etc.) of diffusion models, we collected 200K images from Civitai [10], where images are generated with customized fine-tuned checkpoints, sometimes combined with one or even more adapters.

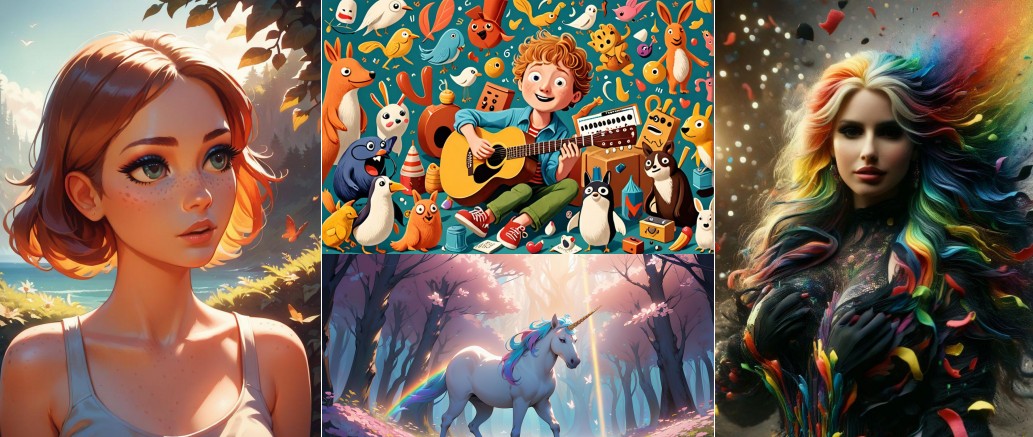

Figure 2: Publicly released images posted in the community. Compared with the images generated directly from the models, the uploader's initial screening ensures quality and better reflects the actual use of the generated images.

Specifically, The prompts used by open-sourced models are selected from the following sources:

- Add additional clauses to each of the labels in ImageNet-21K to ensure photographic style.

- The captions sampled from the stock images as mentioned in real-world images.

- We collected user inputs of the text-to-image service on platform A within a month, filtered out overly simplistic descriptions covered by the first category, and identified 368 most concerned subjects among real users.

- Regarding the most scrutinized issue of portrait forgery, we manually created 63 prompts to generate portrait photos covering various ages, genders, and ethnicities.

As shown in Table 2, our dataset introduces a diversity in style, resolution, provenance, and post-processing that far exceeds that of existing datasets. The analysis of user interaction, on the other hand, makes the distribution of data more consistent with real usage scenarios, making the benchmarks built on this more practical. Details of the sources, resolutions, and number of images are listed in Table 8.

Table 1: Samples of text prompts to generate AI images

| Source | Prompt |
|---|---|
| ImageNet labels | Swiss cheese, highly detailed, HD, 8K, hyperrealistic, full HD, ultra-realistic, high quality, HQ, photorealistic, cinematic lighting, 4K, product photography, ultra detailed, hyper detailed, realistic lighting, sharp focus, hyperrealism |
| Stock Images | Vector illustration of a flying super hero |
| User Input | Fountain pen vector, sticker, artstation, 2d, flat background |
| Human Portrait | Young asian woman, professional photo, raw photo, photo-realistic, best quality, masterpiece, extremely detailed, 2k wallpaper, finely detail, huge filesize, ultra-detailed, highres, extremely detailed, realistic, 8K, ultra-high definition, highest quality, ultra high resolution, high quality texture |

Table 2: Comparison between previous datasets and ACID

| | Previous | ACID |
|---|---|---|
| Source | Generated by a single model | Combinations of models and adapters |
| Content | >90% a single object | Flexible number of objects from 0 to over 5 |
| Style | Photographic / Realistic | Rich Styles |
| Resolution | 50% ≤ 256×256 | 50% ≥ 768×768 |
| Edits | Raw output from models | Include mixed images through augmentation |

### 3.3 AUGMENTATION

Beyond regular on-the-fly image augmentations on single images, to better recognize synthetic images after editing, we also simulated two typical image editing operations within our dataset: image stitching and alpha composition to better recognize synthetic images after editing.

**Image stitching**    This augmentation is designed to simulate the operation of stitching two or more images together. Specifically, we divide an initial image into 16 patches (4x4 grid), randomly fill each patch with either the original image's corresponding area or a matching area from another image randomly selected from the dataset. We consider the resulting stitched image as real-world only if all patches come from real-world images; otherwise, it is considered an AI-created image.

**Alpha composition**    This augmentation aims to simulate the effect of overlaying multiple images. We paste one image over another with a transparency that linearly changes from 1 to 0 either vertically or horizontally. The composite image is considered real-world only if both underlying images are real-world; otherwise, it is categorized as AI-created.

### 3.4 SUMMARY

As demonstrated in Figure 3 and Table 2, compared to previous datasets, the dataset we propose contains higher-quality images, offers a greater diversity of resolutions and aspect ratios, exhibits more significant content variation, and consequently more closely approximates real-world use cases. Additionally, it is straightforward to extend the dataset with future generative models with the list of representative text prompts included.

## 4 ACIDNET

### 4.1 OVERVIEW

As demonstrated in section 5.1, existing solutions fail to achieve satisfactory performance simultaneously on both existing benchmarks and ACID . In concrete terms, methods dedicated to detecting anomalous noise distributions in images like [28, 50, 53, 31, 7] struggle when applied to non-photographic real-world images and images generated by models with distributions that substantially deviate from established ones. On the other hand, methods based on high-level features

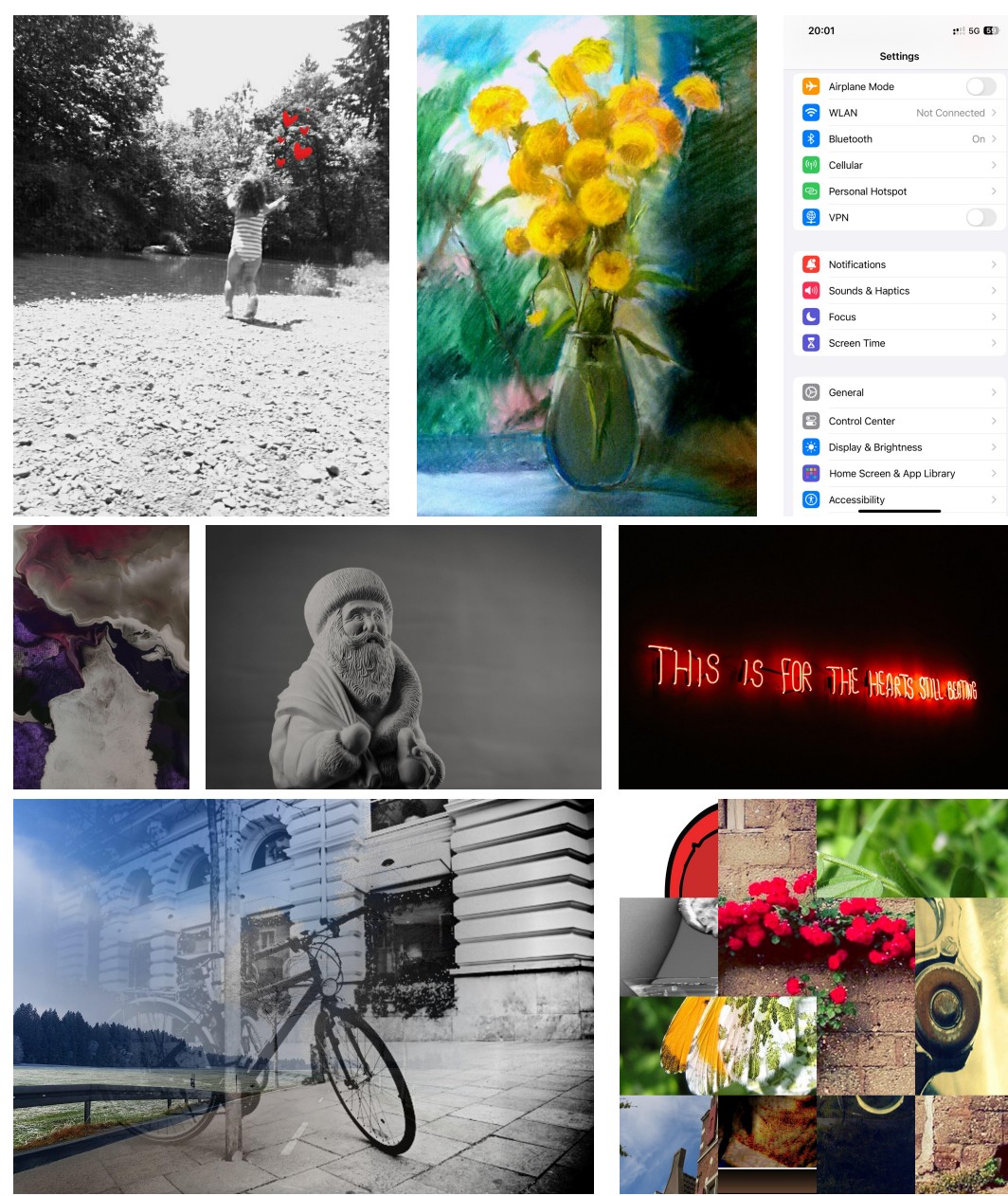

Figure 3: Diversity of real-world images in ACID: the first row shows user images with non-photo elements, the second row illustrates stock images with variant content and styles, and the last row gives examples of augmented real-world images.

like [52, 20] also struggle to generalize across unknown styles and content. Given the limitations of existing solutions, we propose a new baseline method ACIDNet that achieves superior overall performance by simultaneously leveraging both low-level texture and high-level semantic features.

Inspired by [20], ACIDNet consists of two branches, the semantic branch and the texture branch. The semantic branch uses a ResNeXt50 [55] model to extract global semantic features from the original input. Meanwhile, inspired by [7, 59], the texture branch takes the patch with the lowest pixel fluctuation degree, uses a set of high-pass filters proposed in [16] to extract its noise patterns, and obtain texture features with another ResNeXt50 model. Eventually, the texture and semantic features are concatenated and passed through a fully connected layer to make final prediction.

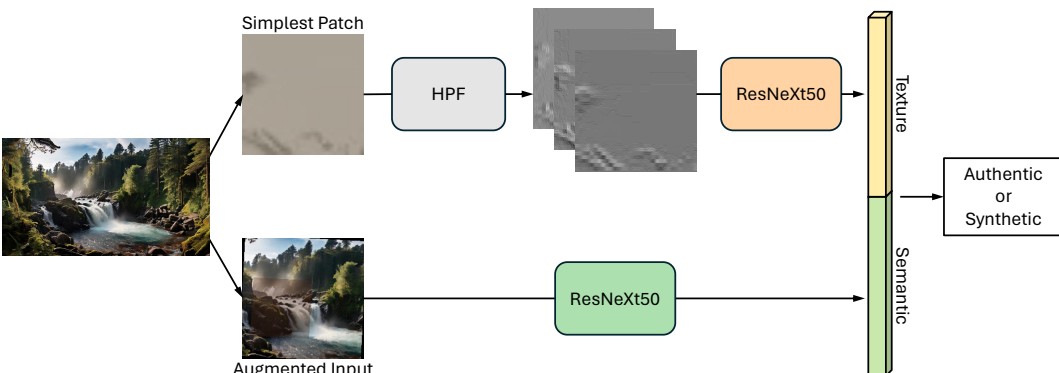

Figure 4: The overall architecture of our method, which consists of three parts: (1) find the simplest patch and extract texture features using high-pass filters (HPF) and ResNeXt; (2) augment the input image and extract semantic features using ResNeXt; (3) concatenate texture and semantic features and make prediction. The augmented input indicates an example of the "add shape" augmentation.

Table 3: Augmentation techniques used in ACIDNet. Techniques marked with (*) are applied exclusively in the semantic branch, while the remaining are applied to both branches.

| Method | Description |
|---|---|
| JPEG Compress | Images are compressed using JPEG algorithm to a random quality level. |
| Random Crop | Images are scaled such that the shorter side measures 256 pixels, followed by a random crop to get images of 224x224 pixels. |
| Add shape* | Add a rectangle of random size, color, and transparency to a random location of the image. |
| Flip | Images are subject to horizontal or vertical flipping, or both. |
| Sharpness Adjust* | Randomly adjust the sharpness of the image. |
| Rotate* | Images are rotated for a random degree between a certain range. |
| Color Jitter | Randomly change the brightness, contrast, and saturation of the image. |
| Gaussian Blur | Blur the image with a random Gaussian blur kernel. |
| Grayscale* | Convert the RGB image to grayscale. |
| Add Noise* | Add a random Gaussian noise to the image. |

### 4.2 SEMANTIC BRANCH

To enhance the capacity to identify images that might have undergone various modifications before publication and during their dissemination, we have implemented several single-image augmentation techniques as listed in Table 3 together with the two cross-image techniques discussed in section 3.3 to simulate common image edits. All the methods except random crop (which is always applied) are applied randomly with a probability of 0.2 and can be stacked.

### 4.3 TEXTURE BRANCH

As proposed in SSP [7], the noise patterns at the simplest patches differ more significantly between real-world and AI-created images since they are more likely to be neglected by generative models. In ACIDNet, 128 patches of $32{\times}32$ pixels are randomly sampled from the original input image, and the patch with the lowest pixel fluctuation degree proposed by Zhong et al. [59] in Equation 1 is considered the simplest patch. $x_{i,j}$ denotes the pixel value of patch $x$ at location $(i, j)$. This metric reflects the complexity of the patches by accumulating the difference between adjacent pixels.

$$L_{div} = \sum_{i=1}^{M} \sum_{j=1}^{M-1} |x_{i,j} - x_{i,j+1}| + \sum_{i=1}^{M-1} \sum_{j=1}^{M} |x_{i,j} - x_{i+1,j}| +$$
$$\sum_{i=1}^{M-1} \sum_{j=1}^{M-1} |x_{i,j} - x_{i+1,j+1}| + \sum_{i=1}^{M-1} \sum_{j=1}^{M-1} |x_{i+1,j} - x_{i,j+1}| \tag{1}$$

Similar to the original input, the simplest patch will be augmented. Different from SSP, the noise fingerprint of the augmented patch is obtained by all 30 high pass filters proposed in [16] rather than 3. The features are passed through a ResNeXt50 model for the texture feature while SSP utilizes ResNet as its backbone.

## 5 EXPERIMENTS

### 5.1 PERFORMANCE ON ACID

To validate the significance of proposing ACID beyond existing datasets, we tested the performance of 7 state-of-the-art pre-trained classifiers on ACID . As shown in the first 7 rows of Table 4, the successes of these classifiers on existing datasets cannot be directly replicated on ACID.

Further experiments indicate that, due to the substantial differences in distributions between the generative models in ACID and some early models from existing datasets, all methods, including ACIDNet , fail to achieve adequate cross-benchmark generalization when trained solely on images from a single source. To determine the capability limits of various solutions, we opt to validate the effectiveness of models on the entire dataset rather than data from a single generative model.

Specifically, after dividing ACID into training and testing sets, we incorporate data generated by ProGAN [21] from the Patchcraft dataset into the ACID training set, and train ACIDNet , SSP [7] and CNNSpot [52] on the combined training data. The reason for selecting these two models is that they presented the best performance in models based on either low-level and high-level features respectively. ACIDNet requires training on 8 Nvidia RTX A6000 GPUs for 3 hours to converge while the other two takes less than 2 hours. The performance listed in the last 3 rows of Table 4 shows that, given adequate amount of training data, ACIDNet achieves a better performance than other models.

Table 4: Performance of different methods on ACID .

| Method | Accuracy | Recall (AI) | Precision (AI) | F1 Score |
|---|---|---|---|---|
| CNNSpot | 52.54% | 2.08% | 12.72% | 3.58% |
| Fusing | 65.65% | 55.53% | 60.20% | 57.77% |
| Gram | 18.70% | 15.34% | 12.49% | 13.77% |
| LGrad | 25.08% | 6.16% | 6.89% | 6.50% |
| UnivFD | 57.55% | 4.91% | 48.36% | 8.92% |
| DIRE (ADM) | 54.07% | 12.65% | 37.35% | 18.89% |
| SSP (ADM) | 58.32% | 1.6% | 90.83% | 3.22% |
| ACIDNet | **98.01**% | 97.05% | **99.47**% | **98.25**% |
| SSP (ACID) | 76.87% | 93.17% | 68.36% | 78.86% |
| CNNSpot (ACID) | 96.74% | **98.67%** | 93.93% | 96.24% |

### 5.2 ABLATION STUDY

To verify the effectiveness of leveraging both high-level semantic features and low-level texture features, we conducted an ablation study by removing either of the two branches and analyzing the performance of two representative models leveraging either of the features. As shown in Table 5, when only low-level texture features are used, the model's performance significantly declines across all datasets; when only high-level features are kept, the model still performs fine on the dataset it was trained, but the generalization capability on other datasets substantially decreases. The composition of PatchCraft [59] and UnivFD [31] are illustrated in Table 7.

### 5.3 HUMAN EVALUATION

Early models or randomly generated images often have obvious flaws (as in Figure 1a), making it difficult for them to cause significant harm even without AI detection techniques. Including AI-generated images that are harder for humans to recognize means that the models trained on such datasets will better help humans avoid being deceived by AI images, thus making them more practically valuable. To test the ability of ACID and previous datasets to deceive humans, we invited

Table 5: Ablation study

| Method | ACID | PatchCraft | UnivFD | Average |
|---|---|---|---|---|
| ACIDNet | 98.01% | **82.82%** | **79.47%** | **86.77%** |
| Texture branch only | 80.14% | 71.79% | 62.78% | 71.57% |
| Semantic branch only | **98.34%** | 65.44% | 70.81% | 78.19% |
| SSP (ACID) | 76.87% | 81.94% | 67.76% | 75.52% |
| CNNSpot (ACID) | 96.74% | 68.09% | 65.37% | 76.73% |

10 individuals to manually distinguish between authentic and AI-generated images. A total of 1500 images (250 AI-generated and 250 real from each dataset) were sampled, randomly shuffled, and resized to 256×256 to avoid resolution bias. As shown in Table 6, ACID proved more challenging for humans to identify.

Table 6: Accuracy in identifying authentic/synthetic images across different datasets by humans.

| | ACID | PatchCraft | UnivFD |
|---|---|---|---|
| Accuracy | **69.83%** | 75.98% | 74.80% |

## 6 LIMITATIONS

In terms of dataset construction, our proposed method necessitates generating thousands of images for each model to assess its detectability, which poses scalability challenges for proprietary models. Regarding the baseline model, although our model shows superior performance with ample training data, its ability to generalize from limited data still requires enhancement.

## 7 CONCLUSION

We introduce a comprehensive new dataset, ACID, which features 13M samples derived from over 50 different diffusion-based models as well as real-world scenarios. The AI-created images in this benchmark are crafted using finely tuned text prompts and span multiple resolutions. For real-world samples, we conducted an extensive search of public data sources, meticulously selecting text-image pairs based on the quality of both visuals and captions.

Leveraging ACID, we developed ACIDNet, an innovative framework to detect AI-created images. ACIDNet combines texture analysis using a simplest patch branch and semantic extraction via a ResNeXt50 branch. With training on more than 10 million 512x512 images from ACID , ACIDNet has achieved an impressive average accuracy of 86.77% in cross-benchmark testing, surpassing previous methods such as SSP and CNNSpot by over 10%.

We plan to openly release both our model and the benchmark dataset to the public. Our goal is to contribute to the ongoing advancements in privacy and AI ethics, particularly as we move towards the era of Artificial General Intelligence (AGI).

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

# A    COMPARISON BETWEEN ACID AND PREVIOUS DATASET

Table 7: Composition of PatchCraft [59], UnivFD [31], GenImage [61], and ACID. Some of the release dates may be inaccurate due to the updates of the models.

| Generative Model | Date | PatchCraft | UnivFD | GenImage | ACID |
|---|---|---|---|---|---|
| StarGAN [9] | 2017 | ✓ | ✓ | | ✓ |
| CycleGAN [60] | 2017 | ✓ | ✓ | | ✓ |
| CRN [8] | 2017 | | ✓ | | ✓ |
| StyleGAN [22] | 2018 | ✓ | ✓ | | ✓ |
| BigGAN [5] | 2018 | ✓ | ✓ | ✓ | ✓ |
| ProGAN [21] | 2018 | ✓ | ✓ | | ✓ |
| STID [6] | 2018 | | ✓ | | |
| Deepfakes | 2019 | ✓ | | | ✓ |
| IMLE [25] | 2019 | ✓ | | | ✓ |
| StyleGAN2 [23] | 2019 | ✓ | ✓ | | ✓ |
| Whichfaceisreal [54] | 2019 | ✓ | ✓ | | |
| GauGAN [33] | 2019 | ✓ | ✓ | | ✓ |
| SAN [11] | 2019 | | ✓ | | |
| ADM [13] | 2021 | ✓ | | ✓ | ✓ |
| GLIDE [30] | 2021 | ✓ | | ✓ | ✓ |
| DALLE2 [40] | 2021 | ✓ | | | ✓ |
| Stable Diffusion v1.4 [47] | 2022 | | ✓ | ✓ | ✓ |
| Stable Diffusion v1.5 [46] | 2022 | | ✓ | ✓ | ✓ |
| VQDM [18] | 2022 | ✓ | | ✓ | ✓ |
| WUKONG [1] | 2022 | ✓ | | ✓ | ✓ |
| Midjourney V5 [29] | 2023 | ✓ | | ✓ | ✓ |
| Midjourney V6 [29] | 2023 | | | | ✓ |
| DALLE3 [32] | 2023 | | | | ✓ |
| Kandinsky 2.2 [4] | 2023 | | | | ✓ |
| Kandinsky 3 [3] | 2023 | | | | ✓ |
| Realism Engine 1.0 [41] | 2023 | | | | ✓ |
| Realistic Vision 5.1 [2] | 2023 | | | | ✓ |
| Stable Diffusion XL [37] | 2023 | | | | ✓ |
| SDXL-Lightning | 2024 | | | | ✓ |
| Stable Cascade | 2024 | | | | ✓ |
| RealVisXL 4.0 [49] | 2024 | | | | ✓ |
| Civitai [10] | 2024 | | | | ✓ |
| Juggernaut XL V9 [45] | 2024 | | | | ✓ |
| Playground 2.5 [36] | 2024 | | | | ✓ |
| Photo Realistic [35] | 2024 | | | | ✓ |

# B DETAILED COMPOSITION OF ACID

Table 8: Composition of ACID

| Model / Source | Count | Resolutions (width $\times$ height) |
|---|---|---|
| ADM | 6000 | $256 \times 256$ |
| StarGAN | 2000 | $256 \times 256$ |
| Civitai | 200793 | Various |
| CRN | 6382 | $512 \times 256$ |
| CycleGAN | 1320 | $256 \times 256$ |
| Dalle2 | 3575 | $224 \times 224$ |
| Dalle3 | 1002250 | $224 \times 224$ |
| Deepfake | 2698 | $256 \times 256$ |
| GauGAN | 5000 | $256 \times 256$ |
| Glide | 6000 | $256 \times 256$ |
| IMLE | 6382 | $512 \times 256$ |
| Juggernaut XL V9 | 3406 | $1024 \times 1024, 1216 \times 832, 832 \times 1216$ |
| Kandinsky 2.2 | 164148 | $768 \times 768, 1152 \times 648, 648 \times 1152$ |
| Kandinsky 3 | 681210 | $768 \times 768, 1152 \times 648, 648 \times 1152$ |
| Midjourney V5 | 6000 | $1024 \times 1024$ |
| Midjourney V6 | 519962 | $1024 \times 1024$ |
| Playground 2.5 | 681210 | $1024 \times 1024, 1024 \times 576, 576 \times 1024$ |
| Photo Realistic | 681210 | $512 \times 512, 768 \times 432, 432 \times 768$ |
| ProGAN | 18003 | $256 \times 256$ |
| Realism Engine 1.0 | 681210 | $768 \times 768, 1152 \times 648, 648 \times 1152$ |
| Realistic Vision 5.1 | 681210 | $512 \times 512, 768 \times 432, 432 \times 768$ |
| RealVisXL 4.0 | 681210 | $1024 \times 1024, 1024 \times 576, 576 \times 1024$ |
| Stable Diffusion 1.4 | 6000 | $512 \times 512$ |
| Stable Diffusion 1.5 | 8000 | $512 \times 512$ |
| Stable Diffusion XL | 681210 | $1024 \times 1024, 1024 \times 576, 576 \times 1024$ |
| SDXL-Lightning | 681210 | $1024 \times 1024, 1024 \times 576, 576 \times 1024$ |
| Stable Cascade | 681210 | $1024 \times 1024, 1024 \times 768, 768 \times 1024$ |
| StarGAN | 2000 | $256 \times 256$ |
| StyleGAN | 5833 | $256 \times 256$ |
| StyleGAN2 | 2000 | $256 \times 256$ |
| VQDM | 6000 | $256 \times 256$ |
| ImageNet | 348780 | Various |
| COCO | 287360 | Various |
| User images | 1715096 | Various |
| Stock images | 1759947 | Various |