# OpenReview forum: "ACID: A Comprehensive Dataset for AI-Created Image Detection"
_ICLR.cc/2025/Conference — ICLR 2025 Conference Withdrawn Submission_

### Official Review · Reviewer_mkpN · 2024-10-29

**Soundness:** 2
**Presentation:** 2
**Contribution:** 2
**Rating:** 3
**Confidence:** 4

**Summary:**

This paper introduces the ACID dataset, comprising 13 million samples collected from over 50 different generative models and real-world sources, offering a broad range of resolutions. Alongside the dataset, the authors propose ACIDNet, a detection model that combines texture and semantic features. ACIDNet achieves 98.01% accuracy on their dataset, surpassing existing methods (e.g., SSP) by over 10%.

**Strengths:**

This paper constructs a large-scale dataset that includes images generated by a variety of generative models, enhancing the dataset's practicality and broad applicability.

**Weaknesses:**

- The paper exhibits some deficiencies in its writing logic. The transitions between paragraphs are not sufficiently cohesive, and the internal coherence within some paragraphs is lacking.

- When describing the dataset, there is a lack of detailed statistical information about the data distribution, such as the number of generated images from different categories or various generative models.

- The paper lacks comparative analysis with other existing datasets in terms of dataset construction; specifically, it could refer to the relevant practices in the GenImage paper.

**Questions:**

- In Table 3, could you provide the parameter settings or random parameter ranges for the following augmentation methods: JPEG Compression, Add Shape, Sharpness Adjustment, Rotation, Color Jitter, Gaussian Blur, and Add Noise?

- In Appendix 9 of the AEROBLADE paper, it is revealed that the image storage format in the dataset can lead models to learn compression biases, significantly affecting model performance. What is the image format of your dataset? Did you use a unified image storage format?

- In Table 4, the top 7 rows use pretrained models to evaluate the generalization of different models on ACID through inference, while the bottom 3 rows use different methods to train and validate on ACID. Placing these two approaches in the same table can be confusing; I recommend separating them into two tables.

- Currently, generated image detection models are not limited to texture and semantic methods. CNNSpot and SSP are not the best-performing detection models. You might consider adding some baselines (e.g., ResNet50, ViT) and some new detection models: DRCT, AEROBLADE, NPR, RIGID, ZED, Fake-Inversion (the first three are open-source, and the others will be open-sourced).

- In line 127, you state that "ACIDNet consistently achieves an average accuracy of 81.1%." How was the 81.1% figure obtained? I only found possibly related data of 86.77% in Table 5.

- In Table 5, what is the difference between "Texture branch only" and "SSP (ACID)"?

---

### Official Review · Reviewer_EhyV · 2024-10-29

**Soundness:** 2
**Presentation:** 3
**Contribution:** 2
**Rating:** 5
**Confidence:** 5

**Summary:**

This paper proposes a large-scale new dataset called ACID, which consists of 13M samples over 50 different recent generative models and real-world scenarios. The dataset is collected from very recent generative models, such as Stable Diffusion XL, with high resolutions, object categories, and augmentation. Furthermore, the authors propose a baseline for their method termed ACIDNet, which consists of two branches: one semantic branch with ResNetXt50, and a texture branch with high-pass filters for a single simple patch. The experiments on their proposed dataset support their method' effectiveness.

**Strengths:**

* The dataset collects images generated from very recent generative models, which should contribute to the related community.
* The authors consider several different scenarios, such as the art, unnatural forgery, and post-processing photos, which are very interesting and should be discussed in this field.
* The dataset considers many different settings, such as style, and object categories, which is also a issue unaddressed by former datasets.
* The proposed detector baseline is effective for detecting AI-generated images, supported by their experiments.
* The paper is well-written and easy to follow.

**Weaknesses:**

* How will the proposed dataset be effective or contribute to future research? Since the generative models always evolving, there will be countless new models in the future. The ACID dataset is novel enough for now, but how to make sure for the future? I acknowledge the authors should have spent enough time and effort on collecting the dataset, but it is not enough if it is just a work depending on time. Maybe there are more insights this dataset can give for related future work.
* The dataset considers many different scenarios and settings, which is good. Therefore, it is a little confusing to follow all the different settings, category them may be better for reviewers to understand, such as for generalization, for robustness, etc.
* For the proposed detector baseline: the resnet branch is a widely-used baseline for image classification, and the texture branch is based on the SSP and Patchcraft, which underestimate the authors own contributions.

**Questions:**

* The authors claim 13M samples for their ACID dataset, but in line 131, they claim 22M images. I don't know whether it is typo.
* The authors regard images uploaded on online platform A before 2019 as not AI-created in line 215. But why? How can you make sure there is no generated/manipulated images before 2019?
* For the post-processing augmentation, did the authors only employ them for training their ACIDNet? Or they also used them to organize their dataset?
* For the simplest patch method, it is a little strange the most discriminative part of an image is the simplest part, since intuitively the more difficult part should also be more difficult to generate. Can the authors provide any proof for this claim beyond two cited previous work?
* For comparisons in Tab.4, the authors compare on their proposed benchmark and show the superiority. Did the authors try to evaluate on other previous public benchmarks? This should provide more evidence for the performance.
* For Tab.4, did the authors evaluate other detectors by using their pre-trained checkpoints? Or fine-tuning on the proposed datasets? We should make the comparisons as fair as possible.

**Details Of Ethics Concerns:**

* This dataset contains different data sources, the authors should make sure everything is ok for, such as privacy, terms of use.
* It could be better to add some ethical discussion on how the dataset and method could impact the community.

---

### Official Review · Reviewer_gfpY · 2024-11-02

**Soundness:** 2
**Presentation:** 2
**Contribution:** 2
**Rating:** 3
**Confidence:** 5

**Summary:**

This paper introduces a new dataset and dual-flow detection framework aimed at addressing the challenges posed by the proliferation of AI-generated images and their potential negative social impacts, such as the spread of fake news.

**Strengths:**

S1: ACID Dataset: The authors present a comprehensive dataset named ACID, which contains 13 million samples sourced from over 50 different generative models and real-world scenarios. The AI-generated images in ACID are created using fine-grained text prompts, and the real-world samples are carefully selected from public data sources based on visual and caption quality, ensuring a broad representation of different image types.

S2: Extensive testing on various AI detectors demonstrates the challenging nature of the ACID dataset. ACIDNet, in particular, shows impressive accuracy of 98.01% on the ACID benchmark, indicating a substantial advancement in the detection of AI-created images.

**Weaknesses:**

W1: The dataset construction requires generating thousands of images for each model, which poses scalability challenges, especially for proprietary models that may not allow such extensive access.

W2: The framework proposed in this paper is simply a combination, lacking innovation. For example, it combines the addition of filters in SSP with the traditional backbone + classifier approach.

**Questions:**

None

---

### Official Review · Reviewer_E4LX · 2024-11-08

**Soundness:** 2
**Presentation:** 2
**Contribution:** 3
**Rating:** 6
**Confidence:** 4

**Summary:**

This paper is relatively well-motivated as AI-generated image detection is a crucial issue. I also find the evaluations thorough.

**Strengths:**

1.The target issues of the paper are meaningful and worth exploring.
2.The motivation is clear.
3.The paper is easy to follow.

**Weaknesses:**

1.The number of images is small. Only 57693 real images and 42307 fake images. This number of images is smaller than GenImage.

2.GAN-based methods are not included in this benchmark.

3.Do the detectors trained on ACID benchmark perform well on real datasets? For example, the images collected from fake news on the Internet.

**Questions:**

See Weaknesses

---

### Note · Authors · 2024-11-12

I have read and agree with the venue's withdrawal policy on behalf of myself and my co-authors.